# Research on In-Flight Alignment for Micro Inertial Navigation System Based on Changing Acceleration using Exponential Function

**DOI:** 10.3390/mi10010024

**Published:** 2018-12-30

**Authors:** Yun Xu, Tong Zhou

**Affiliations:** 1School of Mechanical Engineering & Automation, Zhejiang Sci-Tech University, Hangzhou 310018, China; 2School of Mechanical Engineering, Nanjing University of Science and Technology, Nanjing 210094, China; zhoutong@njust.ed.cn

**Keywords:** micro inertial navigation system, changing acceleration, exponential function, guided projectile

## Abstract

In order to guarantee the stable flight of a guided projectile, it is difficult to realize in-flight alignment for the micro inertial navigation system (MINS) during its short flight time. In this paper, a method based on changing acceleration using exponential function is proposed. First, double-vector observations were derived. Then the initial attitude for the guided projectiles was estimated by the regressive quaternion estimation (QUEST) algorithm. Further, the estimated errors were analyzed, and the reason for using the changing acceleration for the in-flight alignment was explained. A simulation and semi-physical experiment was performed to show the effectiveness of the proposed method. The results showed that the initial attitude error for the rolling angle was about 0.35°, the pitch angle was about 0.1° and the heading angle was about 0.6°, in which the initial shooting angle was between 15° and 55°. In future studies, the field experiments will be carried out to test the stability of the proposed in-flight alignment for guided projectiles.

## 1. Introduction

As we know, an initial navigation system (INS) should carry out the initial alignment before its normal work, such as measuring velocity, attitude, and position [1], when the initial attitude will be determined.

The literature shows that many researchers have focused on the initial alignment for INS [2,3,4,5,6], and some researchers have focused on different filtering algorithms for the initial alignment. Wang et al. presented an improved adaptive Kalman filter for alignment in order to promote the robustness and accuracy of the navigation system [7], simulations and experiments demonstrated the proposed method based on the improved adaptive Kalman filter for the initial alignment. Guo et al. proposed an improved adaptive fading filter for the initial alignment [8]. They solved the problem of the disturbance of inaccurate mode and observation noise. Wang et al. applied an improved fifth-degree cubature Kalman filter algorithm for initial alignment [9], in which the singular value decomposition method was used to improve the numerical stability of the proposed fifth-degree Cubature Kalman filter. Maley presented an algorithm using an extended Kalman filter to estimate the full attitude of a spin-stabilized projectile [10]. In order to enable both the process and measurement models of the Kalman filter to be described as linear and slowly time-varying systems, he introduced a new coordinate system. Simulations have compared extended Kalman filter with steady-state Kalman filter to show the performance of the proposed method. However, other reports have focused on the transfer alignment. Zou et al. proposed a nonlinear transfer alignment of distributed position and orientation system (POS) based on adaptive second-order divided difference filter [11]. They solved the problem of the lever-arm deformation in transfer alignment resulting in a nonlinear error and time-varying measurement noise covariance. Han et al. used the transfer alignment to determine the initial attitude for INS [12], in which the error transfer models in launch-point inertial coordinates and local geographical coordinates were established, and the effects of attitude maneuvers were analyzed under these coordinates. According to the transfer alignment results under these coordinates, they pointed out that the “angular velocity + acceleration” match scheme in launch-point inertial coordinates was an effective method that can determine the initial attitude rapidly and accurately for the slave INS in projectiles. Zhu et al. carried out a transfer alignment on a moving base for low-accuracy slave INS in projectiles, which utilized the output information of a high-accuracy master INS in the aircraft [13]. Si et al. proposed a new transfer alignment method based on relative navigation [14], in which they solved the problem of a complicated and unknown wing flexure deformation influencing the traditional transfer alignment accuracy for airborne weapons. According to the aforementioned researches, it is obvious that the initial alignment method based on different filtering algorithms can meet the requirement of high accuracy, but it will take a relatively long time for the initial alignment to influence the responsiveness of the projectile’s weapon system. Thus, it can only be used in long-range missiles with long flight times but is not suitable for the guided projectiles with short flight times.

As launching environment of the guided projectile is complex because of factors such as the high overloading, high shock and high-speed rotation, several precision devices in the projectile should be powered on after it is launched, especially for a micro inertial navigation system (MINS). Therefore, the initial attitude for the MINS in guided projectiles should be determined during its short flight time. Then, it will be largely different from the application of shipborne, aircraft, and other applications [15,16]. In the aforementioned method, transfer alignment can meet the requirement, but it generally needs both the master INS and slave INS. For the limited interior space of a guided projectile, the space allocation for installing of the master INS and slave INS will be difficult, especially when there is a guided actuator and other systems implemented in the guided projectile.

This paper presents an in-flight alignment method based on changing acceleration using an exponential function for guided projectiles. The organization is as follows: First, according to the navigation velocity rate equation, double-vector observations used for the attitude determination were derived. The regressive quaternion estimation (QUEST) algorithm was introduced, in which the initial attitude can be estimated to encode the initial attitude matrix. Second, estimation errors for the initial attitude were analyzed, which led to the method of changing acceleration using the exponential function. Finally, a simulation and semi-physical experiment was carried out to verify the effectiveness of the proposed method.

## 2. Double-vector Observations and Regressive QUEST Algorithm

Suppose *b* is the body frame of the projectile, and *ib* is the initial body frame of the projectile. For the initial time *t* = 0, the initial body frame is coincident with the body frame. Suppose *n* is the navigation frame, and *in* is the initial navigation frame. For the initial time *t* = 0, the initial navigation frame is coincident with the navigation frame. 

The attitude matrix at any time is $C_{n(t)}^{b(t)}$, and it can be written by:(1)$$C_{n(t)}^{b(t)}=C_{ib}^{b(t)}C_{in}^{ib}C_{n(t)}^{in}$$

According to the definition, for the initial time *t* = 0, $C_{ib}^{b(0)}=1$ and $C_{n(0)}^{in}=1$. Then, Equation (1) can be written as:(2)$$C_{n(0)}^{b(0)}=C_{in}^{ib}$$

Clearly, the problem to solve $C_{n(0)}^{b(0)}$ can be changed to solve the attitude matrix $C_{in}^{ib}$. In general, the attitude matrix $C_{in}^{ib}$ can be encoded by two unparalleled vectors measured in the *ib* and *in* frames. Thus it is necessary to obtain two unparalleled vectors in the *ib* and *in* frame.

### 2.1. Double-Vector Observations

As we know, the navigation velocity rate equation is:(3)$${\dot{v}}^{n}=C_{b(t)}^{n(t)}f^{b}-(2\omega_{ie}^{n}+\omega_{en}^{n})\times v^{n}+g^{n}$$

If we substitute (1) into (3), multiply $C_{n(t)}^{in}$ on both side and integrate it during time 0 to *t*, then it can be written as follows:(4)$$\int_{0}^{t}C_{n(t)}^{in}{\dot{v}}^{n}dt=C_{ib}^{in}\int_{0}^{t}C_{b(t)}^{ib}f^{b}dt-\int_{0}^{t}C_{n(t)}^{in}(2\omega_{ie}^{n}+\omega_{en}^{n})\times v^{n}dt+\int_{0}^{t}C_{n(t)}^{in}g^{n}dt$$

The left part of Equation (4) can be calculated by the subsection integral method as: (5)$$\begin{array}{cc} \int_{0}^{t}C_{n(t)}^{in}{\dot{v}}^{n}dt & ={C_{n(t)}^{in}v^{n}|}_{0}^{t}-\int_{0}^{t}{\dot{C}}_{n(t)}^{in}v^{n}dt={C_{n(t)}^{in}v^{n}|}_{0}^{t}-\int_{0}^{t}C_{n(t)}^{in}\omega_{in}^{n}\times v^{n}dt\\ & =C_{n(t)}^{in}v^{n}-v^{n}(0)-\int_{0}^{t}C_{n(t)}^{in}\omega_{in}^{n}\times v^{n}dt \end{array}$$

Combining (4) and (5), it can be reorganized by:(6)$$C_{in}^{ib}V(i)=W(i),$$

Observations of ***V***(*i*) and ***W***(*i*) are the vectors in the *ib* and *in* frames, and are given by:(7)$$\left\{ \begin{array}{l} W(i)=\int_{0}^{t}C_{b(t)}^{ib}f^{b}dt\\ V(i)=C_{n(t)}^{in}v^{n}-v^{n}(0)+\int_{0}^{t}C_{n(t)}^{in}\omega_{ie}^{n}\times v^{n}dt-\int_{0}^{t}C_{n(t)}^{in}g^{n}dt \end{array}\right.,$$

In general, ***V***(*i*) and ***W***(*i*) are difficult to be parallel [17]. Thus, the attitude matrix $C_{in}^{ib}$ can be encoded according to the vectors of ***V***(*i*) and ***W***(*i*).

### 2.2. Regressive QUEST Algorithm

Wahba’s problem is a classical problem for attitude determination based on double-vector observations [18]. References show that there are many new methods to solve the problem of the attitude determination, such as tri-axial attitude determination (TRIAD), Euler-q, QUEST, and some other improved optimization algorithms [19,20,21].

In this paper, the regressive QUEST algorithm is applied to solve the in-flight alignment problem for the guided projectiles. The algorithm flow chart for the estimation is shown in Table 1.

## 3. Error Analysis of the Alignment Algorithm

From Equation (6), we know that if there is no calculation error for the double-vector observations of ***W***(*i*) and ***V***(*i*) at any time, then the true attitude matrix $C_{in}^{ib}$ can be satisfied by:(8)$$W{\left(i\right)}^{T}=V{\left(i\right)}^{T}C_{ib}^{in},$$

However, there are calculation errors for ***W***(*i*) and ***V***(*i*) at any time, and we suppose, $W=\left[\begin{array}{c} W{\left(1\right)}^{T}\\ W{\left(2\right)}^{T}\\ \vdots \\ W{\left(j\right)}^{T} \end{array}\right]$ is the vector matrix of ***W***(*i*) in each period and its calculation error of it is $\delta W=\left[\begin{array}{c} \delta W{\left(1\right)}^{T}\\ \delta W{\left(2\right)}^{T}\\ \vdots \\ \delta W{\left(j\right)}^{T} \end{array}\right]$, and $V=\left[\begin{array}{c} V{\left(1\right)}^{T}\\ V{\left(2\right)}^{T}\\ \vdots \\ V{\left(j\right)}^{T} \end{array}\right]$ is the vector matrix of ***V***(*i*) in each period and its calculation error of it is $\delta V=\left[\begin{array}{c} \delta V{\left(1\right)}^{T}\\ \delta V{\left(2\right)}^{T}\\ \vdots \\ \delta V{\left(j\right)}^{T} \end{array}\right]$. Suppose the true attitude matrix $C_{in}^{ib}$ is ***X***, and its estimation error for it is *δ**X***. According to Equation (8), we can establish the following equation:(9)W+δW=(X+δX)(V+δV)

Obviously, it is a perturbation linear equation. Then we can reorganize Equation (9) by:(10)$$\delta X={\left[{\left(W+\delta W\right)}^{T}V\right]}^{-1}\left[-{\left(W+\delta W\right)}^{T}\delta V\left(X+\delta X\right)+{\left(W+\delta W\right)}^{T}\delta W\right]$$

According to the consistency of the matrix norms, we can get the following from Equations (10) and (9) by:(11)$$\left\|\delta X\right\|\leq \left\|{\left[{\left(W+\delta W\right)}^{T}V\right]}^{-1}\right\|\left\|{\left(W+\delta W\right)}^{T}\right\|\left[\left\|\delta V\right\|\left(\left\|X\right\|+\left\|\delta X\right\|\right)+\left\|\delta W\right\|\right]$$

(12)‖V‖‖X‖≥‖W‖

Then we can substitute Equation (12) into (11), and reorganize it:(13)$$\frac{\left\|\delta X\right\|}{\left\|X\right\|}\leq \frac{\kappa }{\mu }\left(\frac{\left\|\delta V\right\|}{\left\|V\right\|}+\frac{\left\|\delta W\right\|}{\left\|W\right\|}\right)$$
where, *μ* = $1-\kappa \frac{\left\|\delta V\right\|}{\left\|V\right\|}$, *κ* = $\left\|{\left[{\left(W+\delta W\right)}^{T}V\right]}^{-1}\right\|\left\|{\left(W+\delta W\right)}^{T}\right\|\left\|V\right\|$.

From Equation (7) we know that ***V***(*i*) can be obtained by the observations from the global navigation satellite system (GNSS). Thus, it can be exactly calculated and the errors of *δ**V*** can be approximately 0, then Equation (13) can be written by:(14)$$\frac{\left\|\delta X\right\|}{\left\|X\right\|}\leq \kappa \frac{\left\|\delta W\right\|}{\left\|W\right\|}$$

The vector of ***W***(*i*) can be calculated according to Equation (7) by the observations from gyroscopes and accelerometer. Thus, the error of ***X*** will be largely influenced by the parameter of *κ*. If *κ* is relatively small, then the estimation error of ***X*** will be small. Conversely, if *κ* is relatively large, then the small *δ**W*** will cause a great error of ***X***. 

From the aforementioned analysis, in order to improve the estimation precision of ***X***, *κ* should be largely decreased. As for *κ*, the smaller, the better. According to the literature [22], *κ* is suggested to be in the range of 1 to 100.

In the derivation of Equation (13), the calculation of *κ* relates to ***W*** and ***V***. ***W*** consists of the vector ***W***(*i*), which can be calculated by the observations from gyroscopes and accelerometers. For the real application, gyroscopes and accelerometers will measure the angular rate and acceleration information of the guided projectiles. Generally, the flight angular rate and acceleration of the projectile relate to its flight features; thus it cannot be easily changed. Then, the vector matrix ***W***, which consists of ***W***(*i*), cannot be directly changed. As a result, in order to largely decrease *κ*, we should try to change the vector matrix of ***V***. ***V*** consists of the vector ***V***(*i*), in Equation (7), and ***V***(*i*) can be changed to the following form: (15)$$V(i)=\int_{0}^{t}C_{n(t)}^{in}\left({\dot{v}}^{n}+\omega_{ie}^{n}\times v^{n}-g^{n}\right)dt$$

It is obvious that $C_{n(t)}^{in}$, ${\dot{v}}^{n}$, $\omega_{ie}^{n}\times v^{n}$ and $g^{n}$ will influence ***V***(*i*). However, as we know, $g^{n}$ is a constant vector for the gravity acceleration and cannot be changed. $C_{n(t)}^{in}$ and $\omega_{ie}^{n}$ can be calculated by:(16)$$C_{n(t)}^{in}=C_{e(0)}^{in}C_{e(t)}^{e(0)}C_{n(t)}^{e(t)}$$
(17)$$\omega_{ie}^{n}={\left[\begin{array}{ccc} -\frac{V_{N}}{R_{m}} & \omega_{ie}\cos L+\frac{V_{E}}{R_{n}} & \omega_{ie}\sin L+\frac{V_{E}}{R_{n}}\tan L \end{array}\right]}^{T}$$

According to the observations of velocity and position information from GNSS, $C_{n(t)}^{in}$ and $\omega_{ie}^{n}$ can be calculated, which relates to the velocity and position information of the projectile. They cannot be changed directly. Therefore, in order to change the vector matrix of ***V***, we should try to change the vector of ${\dot{v}}^{n}$ or $v^{n}$. $v^{n}$ originates from the integral of ${\dot{v}}^{n}$, thus, changing the vector of ${\dot{v}}^{n}$ will be an effective way to ensure the change of *V*(*i*) in every moment, which means to change the acceleration of the projectile.

Generally, a change in the vector’s magnitude or the direction will lead to a change in the vector. In order to guarantee the stable flight and precision strike capability of the guided projectile, the flight trajectory of the projectile cannot be largely changed (such as S turn, U turn and so on), which means that the direction of the vector cannot be largely changed. Then, trying to change the magnitude of the acceleration significantly is the only way to realize a change of ***V***. 

Fortunately, a change of the magnitude for the acceleration can be realized by the acceleration or deceleration device of the projectile, such as the turbojet engine, deceleration parachute and so on [23,24,25]. In this paper, in order to realize the change of ***V***, the acceleration using the exponential function is applied [26]. A simulation and semi-physical experiment were performed to verify the effectiveness.

## 4. Simulations and Semi-physical Experiment

In the following, we will describe the simulation and semiphysical experiment to verify the in-flight alignment based on changing acceleration using the exponential function. 

### 4.1. Simulation

For the simulation, the trajectory parameters for the guided projectile are generated by the rigid body trajectory model [27]. Some basic parameter information is as follows.

Suppose the initial velocity is 800 m/s for the projectile when it is shot from the cannon at medium latitude 32°, longitude 118°, and the height is 1000 m. Inertial sensors in the MINS are setting as follows: the zero-drift of gyroscopes is 10°/h with the white noise 10°/h, and the zero-bias of accelerometers is 1 mg, with white noise 1 mg. The velocity observation noise output by GNSS is 0.1 m/s, and the position observation noise is 5 m.

The total flight time is 30 s. The initial attitude for the projectile is *γ_0_* = 15°, *θ_0_* = 35°, *ψ_0_* = 45°, and the initial rolling rate is *ω* = 10°/s. Because the shooting angle will influence the alignment result, it is set from 0° to 90°.

As for the climbing trajectory, in the first 15 s after shooting, the changing acceleration using the exponential function is 35exp(−0.09*t*) m/s^2^, and for the next 15 s the acceleration is −25exp(0.08*t*) m/s^2^. After the first 30 s, the projectile will fly in the attacking trajectory using the initial alignment attitude.

According to the derivation of Equation (13), parameter *κ* is calculated for the total flight trajectory, and it is shown in Figure 1. 

Figure 1 shows the total flight trajectory from *t* = 0 s to 30 s, the calculated parameter *κ* can satisfy the requirement in the range of 1 to 100 when the initial shooting angle is not near 90°. As the shooting angle approaches 90°, *κ* will be extremely large, which means that the alignment result will not be effective.

The results for the proposed alignment method using changing acceleration are shown in Figure 2 during the in-flight time *t* = 0 s to 30 s.

In Figure 2, it is evident that the alignment errors for the rolling error, pitch error, and heading error at time *t* = 0 s are extremely large for different initial shooting angles, and as the alignment time increases, these alignment errors trend toward stability quickly. Thus, it can satisfy the requirement of quickness for the in-flight alignment. However, when the initial shooting angle *θ*_0_ is near 90°, the alignment results for the rolling error, pitch error and heading error are not effective, and are extremely high. This is inconsistent with the analysis of *κ*. According to the attitude definition for the navigation, when the shooting angle is 90°, it is not easy to recognize the rolling angle and heading angle. Therefore, we should avoid an initial shooting angle of 90° for the actual application. 

In our research, we focused on the research of guided projectiles. In general, the initial shooting angle for the guided projectile should not exceed than 80°. Then, the problem can be ignored when the initial shooting angle is 90°. For the future research, we will continue to study the alignment method when the initial shooting angle is 90° for some vertical launch vehicle, such as a rocket, intercontinental ballistic missile and so on.

### 4.2. Semiphysical Experiment

For the limited experimental condition, the semiphysical experiment was conducted with a three-axis flight attitude simulation turntable, which can simulate the rotation of the guided projectile. 

The designed micro inertial measurement unit (MIMU) used for the experiment is shown in Figure 3a, and the turntable is shown in Figure 3b.

The MIMU is made up of micro-electro-mechanical system (MEMS), which contains three gyroscopes and three accelerometers. Before the semi-physical simulation was undertaken, we first carried out a calibration to estimate for errors and compensate for them. For this, we assessed the zero-drifts, installation errors and scale factor errors of gyroscopes and zero-biases, installation errors and scale factor errors of accelerometers. After the compensation, the zero-drifts of the three gyroscopes in MIMU are near 10°/h, and their white noise is about 10°/h, and the zero-biases of the three accelerometers in MIMU are about 1 mg, and their white noise is about 1 mg.

We set the parameters in the control system to realize the rotation of the turntable as the trajectory used in the simulation. Then we acquired the outputs of the gyroscopes in the MIMU. We compared the acquired angular rate with the trajectory data which contains the outputs of specific force for the accelerometers, the velocity and position information for the global navigation satellite system (GNSS), and then implemented the calculation flowchart as shown in Table 1. 

The semiphysical experiment was carried out with some special shooting angles, which contains 15°, 30°, 45°, 55°, and 60°. The statistics for the alignment results are shown in Table 2.

As shown in Table 2, when the shooting angle is between 15° and 55°, the alignment errors indicate that the rolling angle is about 0.35°, the pitch angle is about 0.1° and the heading angle is about 0.6°. The semi-physical experiment can satisfy the requirement of high accuracy for the in-flight alignment. 

The simulation and semi-physical experiment demonstrated the effectiveness of the proposed alignment method based on changing acceleration method. In the future, some field experiments will be carried out to further study the in-flight alignment method for MINS in guided projectiles.

## 5. Conclusions

This paper proposes a method based on changing acceleration using an exponential function to overcome the problem of difficulty in realizing in-flight alignment. The double-vector observations were derived according to the navigation velocity rate equation. The regressive QUEST algorithm was then introduced to encode the initial attitude for the alignment. The estimated errors were analyzed to explain the reason for using the changing acceleration for the alignment. A simulation and semi-physical experiment was conducted, which illustrated that the method could meet the requirements of quickness and high accuracy for the alignment respectively.

In order to attack a target quickly and precisely, the observable heading angle will need to be relatively low as the heading maneuver of the guided projectiles is difficult to realize during its trajectory. However, the rolling angle and pitch angle is changing in every moment during its flight, then, the observability of rolling angle and pitch angle will be high. Thus, the in-flight alignment error for heading angle is relatively high, but the alignment errors for pitch error and rolling error are relatively low. To reduce the alignment error in the heading angle, we will further research the alignment method to improve the observability of the heading angle in order to estimate a high alignment precision for heading, rolling and pitch angles. Also, in our future research, we will focus on the further research of some actual field experiment application to demonstrate the effectiveness of the proposed method, which will promote the applicability of this method.

## Figures and Tables

**Figure 1 micromachines-10-00024-f001:**
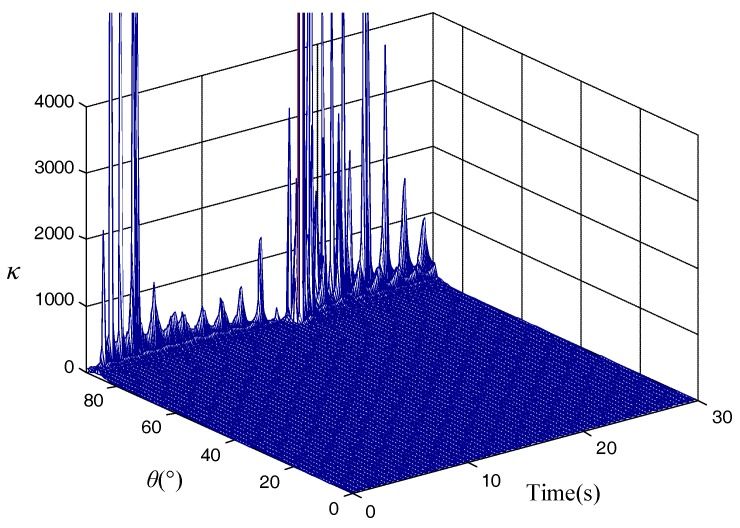
Calculation of parameter *κ*.

**Figure 2 micromachines-10-00024-f002:**
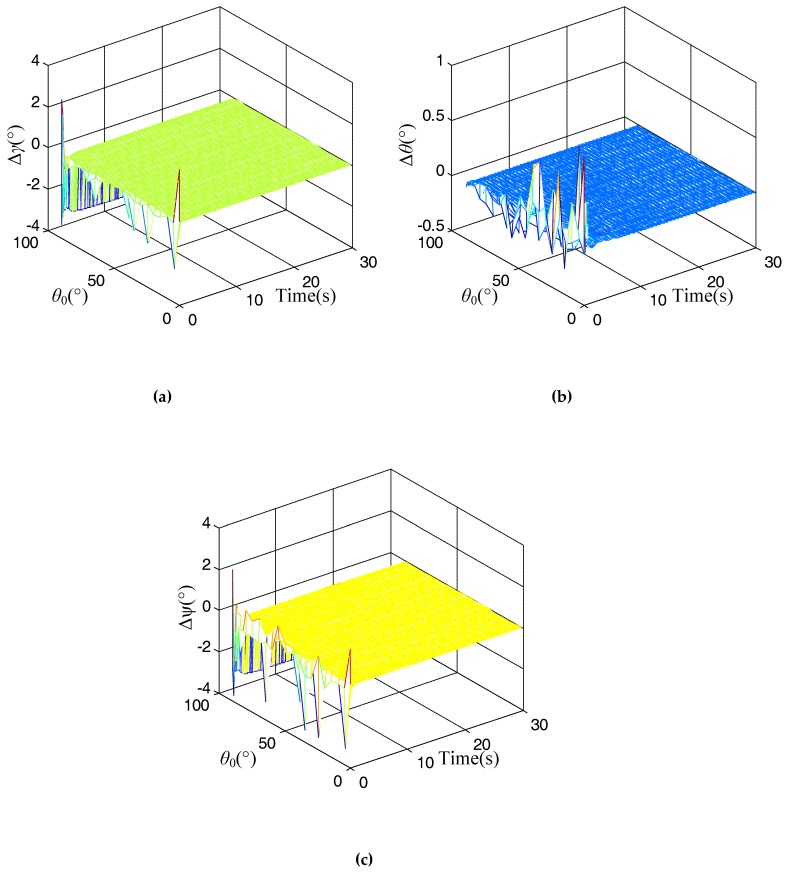
Simulation results for the proposed alignment method. (**a**) Alignment result for rolling error; (**b**) Alignment result for pitch error; (**c**) Alignment result for heading error.

**Figure 3 micromachines-10-00024-f003:**
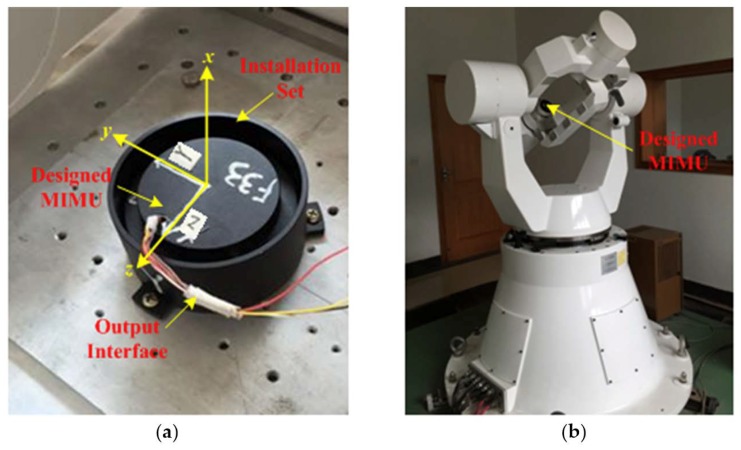
Equipment used for semi-physical experiment. (**a**) The designed micro inertial measurement unit (MIMU); (**b**) three-axis flight attitude simulation turntable.

**Table 1 micromachines-10-00024-t001:** Algorithm flow chart for the estimation of the initial attitude.

Steps	Calculations
1	Initialization, [*q*_0_, *q*_1_, *q*_2_, *q*_3_] = [1 0 0 0]*^T^*, *K*_0_ = 0_4×4_, *m*_0_ = 0
2	In each estimation period, calculate ***W***(*i*) and ***V***(*i*) as shown in Equation (7) and normalize ***W***(*i*) and ***V***(*i*) to be *w*(*i*) and *v*(*i*)
3	Calculate $\delta \sigma =\sum_{i=1}^{j}\alpha_{i}w{\left(i\right)}^{T}v(i)$, $\delta S=\delta B+\delta B^{T}$, $\delta B=\sum_{i=1}^{j}\alpha_{i}w(i)v{\left(i\right)}^{T}$, $\delta Z=\sum_{i=1}^{j}\alpha_{i}w(i)\times v(i)$, $\delta K=\left[\begin{array}{cc} \delta S-\delta \sigma I & \delta Z\\ \delta Z^{T} & \delta \sigma \end{array}\right]$, $\delta m=\sum_{i=1}^{j}\alpha_{i}.$
4	Regressive calculation of $K_{k+1}\;\mathrm{by}:\;K_{k+1}=\frac{m_{k}}{m_{k}+\delta m}K_{k}+\frac{1}{m_{k}+\delta m}\delta K$
5	$\mathrm{Calculate}\;\mathrm{the}\;\mathrm{eigenvectors}\;\mathrm{of}\;K_{k+1}\;\mathrm{and}\;\mathrm{use}\;\mathrm{them}\;\mathrm{to}\;\mathrm{encode}\;\mathrm{the}\;\mathrm{attitude}\;\mathrm{matrix}\;C_{in}^{ib}.$
6	Return to Step 2 for the next period estimation.

**Table 2 micromachines-10-00024-t002:** Alignment results for semi-physical experiment with some special shooting angles.

Shooting angles	Rolling angle error	Pitch angle error	Heading angle error
15°	0.232°	0.106°	0.530°
30°	0.279°	0.112°	0.542°
45°	0.346°	0.114°	0.554°
55°	0.354°	0.113°	0.573°
60°	0.463°	0.113°	0.599°

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
