# Peer review of "Research on In-Flight Alignment for Micro Inertial Navigation System Based on Changing Acceleration using Exponential Function"

_micromachines, 2018, doi:10.3390/mi10010024_

Round 1
Reviewer 1 Report
The innovation points are the analysis of the existing coarse alignment method and the way of improving its performance. In fact, the coarse alignment method mentioned in this paper has been widely applied in engineering, but few researchers give us mathematic analysis about it. My questions are listed as follows:
More references should be added in the introduction section to show how the coarse alignment method develops in the past and now.
Some vectors such as W in (8) should be bold.
Is the regressive quest algorithm analyzed in the section 2?
Author Response
Thank you for your work on my manuscript. The following are the response for your comments or suggestion.
Point 1: More references should be added in the introduction section to show how the coarse alignment method develops in the past and now.
Response 1: According to the reviewer’s suggestion, we have added more references in the introduction section to show the coarse alignment method develops in the past and now. We have discussed that the alignment method can be divided into two kinds. One is based on the filtering algorithm, where we illustrated some literatures to show the research based on different filtering algorithm. For this alignment method, it can meet the requirement of high accuracy, but the disadvantage of it is that it will take a relatively long time for initial alignment, then it will influence the responsiveness of the projectile’s weapon system; the other one is based on the transfer alignment, also, we illustrated some literatures to show the research based on transfer alignment. For this method, it generally needs the master INS and slave INS. For the limited interior space of the guided projectile, the space allocation for the installation of the master INS and slave INS will be difficult. Especially, when there are guided actuator and other implement systems in the guided projectile. The advantages and disadvantages of the initial alignment in the past and now have been compared.
Point 2: Some vectors such as W in (8) should be bold.
Response 2: We have revised the vectors mentioned in this manuscript. Such as W, V , X and so on. Thank you for your suggestion.
Point 3: Is the regressive quest algorithm analyzed in the section 2
Response 3: In section 2, we have analyzed the error of the alignment when using the regressive quest algorithm. Emphatically, the calculating error of W and V has been discussed, which will influence the alignment error of the proposed regressive quest algorithm. Then, it puts forwards to the method based on changing acceleration.
Reviewer 2 Report
The paper is solid. It presents an algorithm that enables a guiding system to locate itself. Although this application is presented as being suitable with the military equipment, different other applications could be identified: UAV, self driving cars, etc.
The paper is presented in clear terms but the language used in the paper is really hard to follow. I suggest a thorough editorial review of the paper. I dare to suggest to have the paper reviewed (editorial issues only) by a native English speaking person or by an English teacher. Lots of the sentences might be understood differently by different readers. Please, do not use abbreviations. Avoid the Saxon Genitive as the paper has under discussion non-life nouns.
Besides, there are few more technical aspects which I will discuss below.
1) Figure 1 and 2 are such cut that the time axis is not cleared defined. This may be the reason that the explanations given to figure 1 and 2 are so vague. I understand that after a short time the error is stabilized. The error is not discussed in the paper next to Figure 2.
2) I also am interested to do what the error will be after feeding the attitude angles as the initial orientation.
3) It may be possible that The caption of Figure 3 should be on same page with the figure 3.
4) I would like to read more details about the experiments – number of repetitions, consistency of the errors. I am sure this is not a problem for the authors as they have, I am sure, extensive results.
5) Meanwhile, some more details about the MIMO block – a schematics or something similar should may increase the quality of the paper.
Author Response
Thank you for your work on my manuscript, I have modified some language which may be hard to follow. Also, we have added the instructions of the abbreviations. The following are the response for your comments or suggestion.
Point 1: Figure 1 and 2 are such cut that the time axis is not cleared defined. This may be the reason that the explanations given to figure 1 and 2 are so vague. I understand that after a short time the error is stabilized. The error is not discussed in the paper next to Figure 2.
Response 1: Thank you for your suggestion about figure 1 and figure 2. We have inserted the relative more clear figures in the manuscript. And we have added the discussion about the errors in the paper next to figure2.
Point 2: I also am interested to do what the error will be after feeding the attitude angles as the initial orientation.
Response 2: Thank you for your suggestion about the interest on the error after feeding the attitude angles as the initial orientation. In this manuscript, as the algorithm flow chart shown in table 1, we supposed that [q0,q1,q2,q3] = [1 0 0 0]T,which means that the initial body frame is coincided with the initial navigation frame. For the future research, we will carried out some further researched on this problem when feeding the attitude angles as the initial orientation.
Point 3: It may be possible that the caption of Figure 3 should be on same page with the figure 3.
Response 3: Thank you for your suggestion about the caption of Figure 3, we have adjusted this caption to be on the same page with Figure 3.
Point 4: I would like to read more details about the experiments – number of repetitions, consistency of the errors. I am sure this is not a problem for the authors as they have, I am sure, extensive results.
Response 4: Thank you for your suggestion. In our real semi-physical experiment, we have carried several numbers of repetitions. And in this manuscript, we calculated the semi-physical experiment data when the shooting angles are 15o, 30o, 45o, 55o, 60o for the discussion. And the statistic results of this semi-physical alignment experiment show in table 2 are based on the calculated results of the aforementioned shooting angles. In the future, we will implement more repetitions and further flight experiments to shown the effectiveness of this method.
Point 5: some more details about the MIMU block – a schematics or something similar should may increase the quality of the paper.
Response 5: Thank you for your
suggestion to add more details about MIMU block. In this manuscript, the
MIMU block is made up of three gyroscopes and three accelerometers. For
the reason that MIMU is constructed by our self, the calibration has
been implemented to compensate the bias errors, the scale factor errors,
the installation errors and so on. The calibration work has been
carried before this work. After the calibration and compensation, the
zero-drifts of three gyroscopes in MIMU are near 10°/h, and the white
noise of them are about 10 °/h, the zero-biases of three accelerometers
in MIMU are about 1mg, and the with white noise of them are about 1mg.
The above instruction has been added in this manuscript.

Round 2
Reviewer 1 Report
It is still no enough technical sound and no proper responds are given to the comments on the first version.
Author Response
In the first version, Reviewer 1 proposed the 3 questions. We have revised the manuscript according to his suggestions one by one. Response to Point 1, we have added more references in the introduction section, which has been tracked in red in the introduction in this revision. Response to Point 2, we have revised the vectors mentioned in this manuscript. Such as W, V, X and so on. Response to Point 3, in section2, we have analyzed the error of the alignment results when using the regressive quest algorithm. The calculating error of W and V has been discussed, which will influence the alignment error of the proposed regressive quest algorithm. Then, it puts forwards to the method based on changing acceleration.
Reviewer 2 Report
As a general observation kindly please have the paper checked by
an English speaker. There are (definite and indefinite) article missing
or some are unnecessary. A very thorough editorial review is necessary
before resubmission.
Line 97 – briefly explain why V(i) and W(i) are difficult to be parallel by one example pls.
Line 107 – the last point of the algorithm (5) – indicate the initial values for this new iteration
Line 122: pls. briefly explain why the error could be estimated as being zero – or provide reference
Line 124 – how the accelerators yield information? Is it by any chance “accelerometers”?
Line 134 and 153 contains conflicting statements – pls. address
Line 141: The statement of line 141 is somehow conflicting with the ones
in lines 139 and 140. I would suggest a line of clarifications
Line 149 – why an exponential function? Explain or reference
If results in Fig 1 are presented between 1 and 50 sec, same duration
should be indicated in Fig. 2 (or reduce the time to 30 sec in fig 1) –
The durations should be consistent in all figures.
Line 196 to 199 – some details about the gyros and the accelerometers
should be provided – given the speciality of the Journal Micromachines,
the specs of the units, the make, the time stability, the interference,
the error angles (Heading error is 600% higher than the pitch angle
error. Why is that? Heading is a very important feature in reaching the
set point and in 50 sec of flight at 600 m/s one could encounter a
significant error. I would suggest to include in the conclusion such a
discussion (10 to 20 lines) and eventually discuss the future work
through the way this problem could be mitigated.
Author Response
We have revised our manuscript according to Reviewer 2's suggestion and the following are the responses. Thank you very much for your work on my manuscript.
Response 1: Line 97 – V(i) and W(i) are difficult to be parallel generally, it can be referred to the reference[17].
Response 2: Line 107 – the initial values for this new iteration is firstly indicated in step 1, where, m0 = 0 and K0 = 04×4.
Response 3: Line 122 – δV means calculation error for the actual V and the calculated V. Suppose, there’s no measured errors for GNSS and no calculation errors, then, vector of V in every moment can be exactly calculated, thus, the error of δV could be approximated to 0.
Response 4: Line 124 – yes, it is a mistake in writing, it should be accelerometers, and we have corrected it.
Response 5: Line 134 and 153 – thank you for your suggestion, we have added some statements.
Response 6: Line 141 – thank you for your suggestion, we have added some explain.
Response 7: Line 149 – generally, according to the rules of guidance and control, the acceleration of the guided projectiles is ν' = – cx·ρ/2·v2·s/m – g·sinθ, where, the model of exponential function is used for ρ , more details of it can be referred to the reference [27].
Response 8: Fig.1 – thank you, we have reduced the time to 30s in fig.1 according to your suggestion.
Response 9: Line 196 to 199 – thank you, we have added the details about the gyros and the accelerometers according to your suggestion. For the reason that the large heading maneuver of the guided projectiles is so difficult to realize during its trajectory, then the observability of heading angle will be relatively low. However, the rolling angle and pitch angle is changing in every moment, then, the observability of rolling angle and pitch angle will be high. Thus, the in-flight alignment error for heading angle is relatively high, but the alignment errors for pitch error and rolling error are relatively low. In order to reduce the alignment error of heading angle, we will further research the alignment method to improve the estimating precision of heading angle error. Also, in our future research, we will focus on the further research of some actual field experiment application to demonstrate the effectiveness of the proposed method, which will promote the applicability of this method. The aforementioned discussion has been added in the revised manuscript.